# Evaluation of Semen Self-Sampling Yield Predictors and CTC Isolation by Multi-Color Flow Cytometry for Liquid Biopsy of Localized Prostate Cancer

**DOI:** 10.3390/cancers15102666

**Published:** 2023-05-09

**Authors:** Cesare Saitta, Ilaria De Simone, Vittorio Fasulo, Marinella Corbetta, Stefano Duga, Chiara Chiereghin, Federico Simone Colombo, Alessio Benetti, Roberto Contieri, Pier Paolo Avolio, Alessandro Uleri, Alberto Saita, Giorgio Ferruccio Guazzoni, Rodolfo Hurle, Piergiuseppe Colombo, Nicolò Maria Buffi, Paolo Casale, Giovanni Lughezzani, Rosanna Asselta, Giulia Soldà, Massimo Lazzeri

**Affiliations:** 1Department of Urology, IRCCS Humanitas Research Hospital, 20089 Rozzano, Italy; 2Department of Biomedical Sciences, Humanitas University, 20072 Pieve Emanuele, Italy; 3Flow Cytometry Core, IRCCS Humanitas Research Hospital, 20089 Rozzano, Italy; 4Department of Pathology, IRCCS Humanitas Research Hospital, 20089 Rozzano, Italy

**Keywords:** prostate cancer, liquid biopsy, circulating tumor cells

## Abstract

**Simple Summary:**

There is a growing trend towards exploring the use of non-invasive liquid biopsy (LB) for prostate cancer (PCa) detection. The primary objective of this study is to identify the rate of patients who accepted and were able to collect seminal fluid. The secondary objective was to evaluate the efficiency of our protocol in collecting prostate-derived cells. Our study shows that only one third (139/356) of patients affected by locally advanced PCa, who accepted to collect their seminal fluid, were able to donate. The main favorable predictors for semen collection were young age and lower prostate volume. Putative cancer cells (PSMA + EpCAM^high^) were 2% of the isolated cells in urine and semen. The fraction of EpCAM^high^ cells over prostate-enriched cells (PSMA+) significantly correlated with patient age in both semen and urine, but not with other clinical parameters.

**Abstract:**

Liquid biopsy (LB) for prostate cancer (PCa) detection could represent an alternative to biopsy. Seminal fluid (SF) is a source of PCa-specific biomarkers, as 40% of ejaculate derives from the prostate. We tested the feasibility of an SF-based LB by evaluating the yield of semen self-sampling in a cohort of >750 patients with clinically localized PCa. The overall SF collection yield was 18.2% (39% when considering only compliant patients), with about a half of the patients (53.15%) not consenting to SF donation. Independent favorable predictors for SF collection were younger age and lower prostate volume. We implemented a protocol to enrich prostate-derived cells by multi-color flow cytometry and applied it on SF and urine samples from 100 patients. The number of prostate-enriched cells (SYTO-16+ PSMA+ CD45−) was variable, with higher numbers of cells isolated from SF than urine (*p* value < 0.001). Putative cancer cells (EpCAM^high^) were 2% of isolated cells in both specimens. The fraction of EpCAM^high^ cells over prostate-enriched cells (PSMA+) significantly correlated with patient age in both semen and urine, but not with other clinical parameters, such as Gleason Score, ISUP, or TNM stage. Hence, enumeration of prostate-derived cells is not sufficient to guide PCa diagnosis; additional molecular analyses to detect patient-specific cancer lesions will be needed.

## 1. Introduction

Prostate cancer (PCa) accounts for 26% of all incident cases in men, with 268,490 new cases in 2022 in the United States only [1]. Diagnosis of PCa begins with a digital rectal examination (DRE) and prostate specific antigen (PSA) measurement, complemented by imaging, such as multiparameter magnetic resonance imaging (mpMRI), and is finally confirmed by trans-rectal or trans-perineal prostate biopsy. Biopsy may result in patient discomfort (pain), morbidities (rectal bleeding, gross hematuria, hemospermia, local infection, sepsis, and death), and increased costs for the health system [2]. As an alternative to standard invasive diagnostic procedures, in the last decade, there has been a growing interest in circulating tumor cells (CTCs) and circulating cell-free tumor nucleic acids (ctNAs), which are released in the blood and other bodily fluids by primary or metastatic tumors. They have been considered a non-invasive source for “liquid biopsy”, ensuring a personalized tool for cancer detection and characterization [3]. Evidence exists for the detection, enumeration, and molecular characterization of CTCs and identification of ctNAs for different cancers, with their counts being helpful and clinically relevant, especially in patients with metastatic diseases, including PCa. However, those markers are below the threshold for detection in localized PCa, even in high-risk patients [4]. Consequently, blood liquid-biopsy for early diagnosis of PCa has shown several drawbacks, limiting its clinical utility [5]. For instance, the first studies demonstrated that CTCs in localized PCa were detected in a lower number of patients (between 8% and 27%) compared to metastatic castration resistant PCa. In addition, the presence and number of CTCs did not correlate with other clinicopathological parameters, such as Gleason Score (GS), serum PSA, and pTNM stage [6]. Besides blood, SF and urine are much less explored as a source of CTCs, despite the longstanding evidence that both might contain prostate-derived tumor cells [7]. Considering that the prostate gland contributes approximately 30–40% of seminal fluid (SF) and that smooth muscle contraction ejects the prostate material into the urethra upon ejaculation, SF could be an ideal liquid for a non-invasive early detection of PCa. Besides few attempts to isolate CTCs from semen [7,8], in 2018, we showed preliminary results indicating that PCa cells can be retrieved from semen and cancer-specific genetic alterations may be efficiently detected starting from a heterogeneous cell population [9]. In addition, a novel method based on microfluidic separation was recently reported [10]. However, SF collection may present several drawbacks due to age, erectile dysfunction, drugs (i.e., alpha-blockers), personal inhibition, and values. Here, from one side, we investigated predictors of SF collection in men with clinically localized PCa, and from the other, we tested a novel protocol for CTC isolation from semen based on multi-color flow cytometry.

## 2. Materials and Methods

### 2.1. Patient Cohort

Patients were initially recruited through an institutional, spontaneous, observational, longitudinal, prospective study started in 2017 (Prot. ICH-N. 336/19, 14 May 2019), which was later incorporated into an Italian Ministry of Health granted project (Ricerca Finalizzata 2018, grant number RF-2018-12367080). Subjects were men with a diagnosis of clinically localized PCa previously subjected to systematic biopsy, and/or software assisted imaging-guided fusion trans-rectal or trans-perineal prostate biopsy, and scheduled for whole gland treatments (radical prostatectomy, RP, or radiotherapy), according to the European Association of Urology guidelines for organ confined PCa [11]. Both studies were designed according to the STARD statement and were approved by the Ethical committee of the IRCCS Humanitas Research Hospital. The inclusion criteria were the following: (i) age 45–75 years; (ii) positive biopsy for adenocarcinoma of prostate according to the International Society of Urological Pathology grade (ISUP) [12,13]; (iii) imaging diagnosis of organ confined PCa; and (iv) Eastern Cooperative Oncology Group (ECOG) performance status 0–1. The exclusion criteria were the following: (i) patients with bacterial acute prostatitis in the 3 months prior to treatment; (ii) patients subjected to previous endoscopic surgery of the prostate; (iii) patients subjected to pelvic radiotherapy; and (iv) patients with neurological illness impacting the autonomic system. All patients were contacted and visited by a senior urologist (A.B.), trained for physician–patient communication, who informed them of the details and aims of the project. Patients who accepted signed an informed consent. The primary outcome of the study was to report the rate of men who consented to collect the SF, calculated as the ratio between all the patients contacted and invited over those who consented. The secondary outcomes were: (i) to report the rate of men who collected the SF, calculated as the ratio between those who consented over those who were able to collect SF by self-ejaculation; (ii) to report variables associated to semen collection or failure to collect; and (iii) to report the rate of men for whom both SF and urine were available for subsequent analyses of circulating prostate-enriched cells.

### 2.2. Sample Collection and Pre-Processing

At the time of admission to the hospital, usually the day before surgery or radiotherapy, each participating patient was asked to provide a SF and a post-ejaculation urine sample. Patients were invited to collect the seminal and urine samples at home, using two different sterile collection vessels provided by the lab or pharmacy, and return the samples within 2 h from collection. They should not have sexual activity for 2 to 5 days before collecting the sample. All patients were instructed to collect the sample by cleaning the head of the penis with wet, soapy wipes or cotton balls and drying the glans well before collecting the sample. The sample was obtained by self-masturbation, and not during intercourse. The volumes of SF samples ranged between 0.1 and 5 mL (median 1.5 mL). The patients were also asked to collect a urine sample after ejaculation. The volumes of urine samples ranged between 3 and 30 mL (median 25.5 mL). Both semen and urine samples were centrifuged at 450× *g* at room temperature for 10 min to separate the cellular component from the seminal plasma. In case the SF was particularly rich in sperms, the sample was subjected to a step of liquefaction at 37 °C with gentle mixing for 30 min before centrifugation. The cellular component was further processed by FACS to enrich for prostate-derived circulating cells.

### 2.3. FACS Sorting 

The protocol for the recovery of prostate-derived cells from SF was set up using a spike-in approach, as previously described [9]. Briefly, semen samples from healthy donors were spiked with different amount (10, 100, 1000) of LNCaP cells, a human prostate adenocarcinoma cell line with a well-defined 1-bp insertion within exon 9 of the *JAK1* gene (NM_002227.3: c.1282_1283insC). After cell-sorting, the actual presence of LNCaP cells was verified by a competitive fluorescent PCR assay (Appendix A). The estimated recovery of LNCaP cells was 70.4% ± 10.8% [9]. As control, ejaculate samples from 6 healthy donors were obtained and processed to evaluate basal levels of prostate-derived cells in semen.

The cellular component from SF and urine was processed to be analyzed by the FACS Melody sorter instrument (BD Biosciences, San Jose, CA, USA). Briefly, pellets were washed with RPMI 1640 Complete Medium (Euroclone, Wetherby, UK), supplemented with 2 mM L-Glutamine and 10% Fetal Bovine Serum (FBS; VWR International, Edison, NJ, USA), and centrifuged at 450× *g* at room temperature for 10 min. Then, cells were labelled with the following nucleic acid fluorescent dyes and antibodies: 7-Amino-actinomycin D (7-AAD, BD Pharmingen, San Diego, CA, USA), SYTO-16 (Thermo Fisher Scientific, Waltham, MA, USA), CD45 (BD Pharmingen, San Diego, CA, USA), PSMA (BioLegend, San Diego, CA, USA), and EpCAM (BD Biosciences, San Jose, CA, USA). The conjugated fluorophores and the units are summed up in Table 1. After 15 min of incubation at room temperature in the dark, the mixture was washed with a solution of 1× PBS (Phosphate Buffered Saline) with 1% FBS and centrifuged at 450× *g* for 5 min to remove the unbound antibodies. Pellets were diluted with the same solution and loaded into the FACS Melody sorter instrument. Cells were sorted in 300 μL of PBS. 

### 2.4. Statistical Analyses 

Data were analyzed using either the STATA 17/SE package software (StataCorp, College Station, TX, USA) or R version 4.0.3 (RStudio Team 2020; http://www.rstudio.com/ (accessed on 31 March 2023)). The normality Kolmogorov–Smirnov test was used to check the distributions of the data. The descriptive part of the analysis was carried out by computing absolute frequencies and percentages for the qualitative (categorical) variables. For the quantitative analysis, mean and standard deviation were computed if the variable under consideration was normally distributed, while median and interquartile range (IQR) were used for variables not normally distributed. For continuous variables, the t-Student paired-test was used for normally distributed data, while the Wilcoxon signed-rank test was used for non-normal data. The chi squared test with Yates correction or the exact Fisher test were adopted for categorical variables. For quantitative variables instead, the t-Student test for unpaired data and the U-test of Mann–Whitney were used, respectively, depending on whether the variable followed or not a normal distribution. A multivariate logistic regression model was fitted for the outcome of interest through a stepwise approach choosing the covariates with the best prediction performances. The accuracy of the model was determined as the area under the receiver operator (ROC curve). Finally, statistical significance of enriched prostate-derived cells fractions was calculated by the Mann–Whitney test. Data transformation was applied to those values of interest (numbers of cells and numbers of cells/mL) that ranged over several orders of magnitude and whose distribution significantly deviated from normality. Specifically, we applied a logarithmic transformation (log10) and verified, with the Kolmogorov–Smirnov test, that normality has been achieved after the transformation. Statistical significance of sorted cells across risk class was calculated by one-way ANOVA test. Differences leading to a *p*-value ≤ 0.05 were considered statistically significant. In case the test was statistically significant, a Pearson correlation coefficient (r) was also calculated to determine whether the variables were positively or negatively (anti-) correlated.

## 3. Results

### 3.1. Predictors of Semen Self-Sampling Yield 

Before addressing whether the liquid biopsy of SF is a valid and clinically useful option, we evaluated the patients’ willingness to participate and provide a SF sample. From July 2017 to December 2021, 760 patients with localized PCa, who were scheduled for whole gland treatment (RP or radiotherapy), were proposed to enter the study. Patients’ demographic and clinical characteristics are shown in Table 2. PCa characteristics are described by multiple classification systems, including: (1) the PIRADS (Prostate Imaging Reporting And Data System) score, which is based on multiparametric MRI findings and ranges from 1 to 5, where higher scores indicate a progressively increased probability of having a clinically significant PCa; (2) the ISUP histopathological grading, which defines 5 grade groups based on GS (ISUP 1: GS ≤ 6; ISUP 2: GS 3 + 4; ISUP 3: GS 4 + 3; ISUP 4: GS 8; ISUP 5: GS ≥ 9); (3) the pathological Tumor, Node, Metastasis (pTNM) staging, which gives an idea of the anatomical extension of the tumor within the prostate (pT2, PCa confined within the prostate, pT3, extracapsular extension or seminal vesicles invasion) and in the regional lymph nodes (pN); and (4) the positive margins, referring to the presence/absence of cancer cells at the edge of the tissue removed by surgery. In the entire cohort, mean age was 65.5 (±SD 8.1) years, median pre-biopsy PSA was 7.01 ng/mL (IQR 5.1–11.6). Family history (limited to father and brothers) was found in 85 (11%) patients, mpMRI was performed in 461 patients before biopsy. The median biopsy ISUP was 2 (IQR 1–3) and the median ISUP at the final pathological report was 2 (IQR 2–3). RP was performed in 279 patients, while 481 opted for alternative non-surgical treatments. The PIRADS score was determined in 53.9% of the patients. Among the patients with available data, 4.3% had a PIRADS score of 1, 0.1% had a score of 2, 12.5% had a score of 3, 26.4% had a score of 4, and 10.5% had a score of 5. The median prostate volume at the time of MRI was 50 cc, with a IQR of 36–70 cc. Considering only the patients who underwent surgery (279), at the final histological report, most patients had an ISUP grade of 2 (127, 45.5%), followed by grades 3 (78, 28.0%) and 5 (36, 12.9%). A smaller percentage of patients had grades 1 (16, 5.7%) and 4 (22, 7.9%). No statistically significant difference was recorded between biopsy ISUP and ISUP at the final histological report (*p* > 0.05). The predominant pT stage was pT2c (148, 53.0%), while stages pT3a and pT3b accounted for 25.4% (71) and 10.4% (29) of patients, respectively. Stage pT2a and pT2b were less frequent (10.4% and 0.7%, respectively). Regarding lymph node involvement, most patients (251, 90.0%) had no evidence of pN involvement (pN0), while 10.0% (28) had evidence of PCa having spread to local (pelvic) lymph nodes (pN1). Similarly, 81.7% (228) of the patients had negative margins (R0) and 18.3% (51) had positive margins (R1). The percentage of tumor involvement varied across patients, with a median value of 15% and a IQR of 8–20%. Of 760 patients, 404 (53.15%) declined to participate (Figure 1). As they did not sign any informed consent and rejected to participate to the study, no further information about the reasons why they declined to collect SF is available. Among the 356 patients who accepted to participate, 139 (39%) were able to collect a SF sample (Figure 1). Considering 217 patients who did not collected the SF, 130 (59.9%) reported that failure was due to psychogenic inhibition at the time of masturbation, 33 (15.2%) suffered from erectile dysfunction, and 54 (24.9%) had retrograde ejaculation due to alpha blocker treatment. Considering the total amount of patients who were able to collect semen (139), in most of the cases (100, 72%), urine was also collected. Table 2 also shows the baseline differences between patients who decided to participate and those who declined to participate in the project. Pathological data from patients who did not consent to donate their SF, but underwent RP, were retrospectively reviewed and used for Table 2. As shown in Table 2, patients who decided to participate had higher total PSA level, PIRADS at mpMRI, higher percentage of tumor and ISUP at the final histological report, and a lower prostate volume in a statistically significant manner. Moreover, when stratified according to the ability of semen collection, as illustrated in Table 3, only age and prostate volume demonstrated statistically significant differences between the two groups (*p* < 0.001). Finally, we fitted a logistic regression model to test the independent predictors of semen collection. In the univariate and multivariate analysis, age (OR 0.95; CI 0.91–1.0; *p* < 0.05) and lower prostate volume (OR 0.97; CI 0.95–0.98; *p* < 0.001) were the only independent predictors of semen collection (Table 4), with an area under the receiver operator of 0.73 (Figure 2). 

### 3.2. Isolation of Prostate-Derived Cells from SF and Urine by FACS

We set up a protocol to enrich prostate-derived cells from SF and urine (Figure 3) and evaluated whether their number correlates with clinical parameters. The analysis was performed on a total of 100 patients, for whom both SF and post-ejaculation urine samples were available. Of these, two SF and five urine samples had sub-optimal quality and were not processed by FACS. To isolate and enrich prostate-derived circulating cells, we first separated the cellular component by centrifugation and then labelled samples with a cocktail of two nuclear dyes and three antibodies for multi-color FACS analysis (Table 1). Samples were then processed one at a time on a Melody FACS instrument (BD Biosciences) according to the gate strategy shown in Figure 3. First, cells viability was assessed using the DNA binding markers 7-AAD and SYTO-16; while SYTO-16 is a cell-permeable marker and enters cells with an intact membrane (typically live cells), 7-AAD is a cell-impermeable marker that binds nuclei of apoptotic cells [14,15]. Then, live cells (P1, e.g., SYTO-16+ and 7-AAD−) were characterized for their expression of the PSMA and the CD45 markers, which label prostate cells and immune cells, respectively. All prostate-derived cells (SYTO-16+, PSMA+, CD45−), occurring in P2 gate were sorted and collected for downstream analyses. This cell fraction likely also contains tumor-derived cells. To estimate the number of CTCs within the collected cells, P2 gate cells were further analyzed according to EpCAM expression combined with PSMA. All the events occurring in P3 gate (high EpCAM coupled with PSMA positivity) could represent putative CTCs. Data obtained on healthy donors (*n* = 6) showed that the basal number of putative CTC (SYTO16+ PSMA+ CD45− EpCAM^high^) ranged between 3 and 17.7 cell/mL [9], while prostate-derived cells (P2 fraction, SYTO16+, PSMA+, CD45−) ranged between 177 and 404 cells/mL.

### 3.3. Enumeration of Prostate Derived Cells in SF and Correlation with Clinical Parameters

The number of sorted cells (P2: SYTO-16+, PSMA+, CD45−) was highly variable and likely depended on different factors, such as patients’ state of health, starting volume, and composition of liquid biopsies. In our experiment, we collected, on average, 54,802 (range 40–324,033) and 1143 (range 16–8830) prostate-derived cells per mL in SF and urine, respectively. The estimated number of cancer-derived cells (P3: SYTO-16+, PSMA+, CD45−, EpCAM^high^) per mL was, on average, 13,519 (range 28–104,339) in SF and 430 (range 2–3210) in urine. Hence, on average, prostate-derived cells isolated from SF are significantly more numerous than those enriched from urine (*p* value < 0.001, Mann–Whitney test). To compare cell subpopulations characterized by FACS while reducing the effect of inter-sample variability, we considered the fraction of occurrences in each gate out of the total population or gated-parent population (Figure 4 and Appendix A). In this case, we observed that the percentage of live cells (P1) out of total population was, on average, higher in SF than urine (49% and 34%), and the percentage of sorted cells (P2) was, on average, 6.4% and 4.7% for semen and urine, respectively. Instead, the fraction of immune cells on all live cells (CD45/P1) and the fraction of EpCAM^high^ cells in all sorted events (P3/P2) are significantly higher in urine than in semen. However, the fraction of putative tumor- derived cells (P3) on total population does not show consistent differences between the two sources (1.6% and 1.8%, on average, in semen and urine, respectively) (Figure 4). 

Finally, we evaluated the potential diagnostic and prognostic applicability of the presented method by correlating selected clinical parameters, such as pre-biopsy serum PSA level, Gleason Score, ISUP, TNM stage, prostate volume, percentage of tumor over the entire gland, and age, with each of the following (Appendix A): (i) the absolute number of P2 (SYTO-16+, PSMA+, CD45−) prostate-derived cells (log10(*n*)), (ii) the number of prostate-derived cells per mL of the starting sample (log10(P2/V)); (iii) the estimated P3 (SYTO-16+, PSMA+, CD45−, EpCAM^high^) number (log10(P3/V)); (iv) the estimated number of P3 per mL of the starting sample (log10(P3/V)); and (v) the fraction of EpCAM^high^ cells over prostate-enriched cells (P3/P2). Age was the only clinical parameter consistently and positively correlated with the fraction of EpCAM^high^ cells over prostate-enriched cells (PSMA+) (Figure 5A) in both SF (*p*-value = 8.16 × 10^−7^) and urine (*p*-value = 0.00108), whereas prostate volume correlated with the concentration of EpCAM^high^ cells/mL (Figure 5B) in urine (*p*-value = 0.0184) but not in semen.

## 4. Discussion

Despite a variety of approaches that have been developed to identify CTCs, primarily from blood, their use as a biomarker for localized and locally advanced PCa has been limited. The main challenges concern the rarity of CTC in blood at early PCa stages, their phenotypic heterogeneity, and the lack of standardized methods for the sensitive and specific detection of prostate-derived cancer cells. Even the CellSearch^®^ method, which is currently FDA-approved for the enumeration of CTC in metastatic PCa, provided variable results and limited clinical value when applied to non-metastatic disease [16,17]. 

Semen and urine represent the most obvious alternatives to obtain diagnostically relevant amounts of cancer cells from PCa patients [17,18] due to both the anatomical and functional features of the prostate, which is connected to the lower part of the urinary tract and directly secretes prostatic fluid into the urethra during ejaculation. Therefore, it is expected that prostate-derived cells would be naturally released in semen (and urine), allowing them to be retrieved more easily and in larger numbers than in blood. Moreover, since the shedding of cells from prostate to SF and urine does not require disruption of the tissue–blood barrier and intravasation, these biofluids seem promising for earlier detection of cancerous cells. Finally, another main advantage of liquid biopsy of both semen and urine over blood is the possibility of patient self-collection without the need of a skilled medical professional, making them truly non-invasive. In the case of urine, the sample volume is not a limiting factor, and the storage and transport of the sample are easy [17]. However, sample volumes are more variable across patients and due to the large volumes, overall CTC concentration may be low. Instead, semen volumes are more homogeneous among patients and the ejaculate is known to be enriched in prostate-derived elements, which are found at a much higher concentration than in urine or blood. Indeed, PSA was initially identified in semen [19].

Although the SF liquid biopsy for PCa detection could be considered attractive, urologists should address the real-life scenario and consider the self-SF collection yield before starting with specific research projects aiming to use CTCs and/or cfNAs as an alternative to invasive prostate biopsy. In our population of patients with PCa who are scheduled for treatment, the overall SF self-collection yield was low (18.2%, 139/760). In patients who consented to participate in the study, the rate of seminal fluid sampling was 39%. Possibly, the knowledge of the disease and the scheduled treatment options may have impacted the willingness to participate and donate. However, compliance may be higher in a different group of patients (for example younger people, or people that still do not have a diagnosis) and with accumulating evidence in the future that liquid biopsy of semen can indeed provide an advantage to PCa patients. Regardless of the treatment, anxiety and depression are the most common psychological conditions affecting cancer patients, leading to a significant deterioration in QoL [20]. The patients may experience feelings of anxiety and distress while living with “untreated” cancer waiting for treatment, which could be the main reason to not participate in the study or be inhibited in collecting [21], although we did not specifically assess the patients with a questionnaire, such as the General Anxiety Disorder scale 7 (GAD-7), the Distress Thermometer, or the Patient Health Questionnaire PHQ-9 for depressive symptoms. Age may be one of the other major drawbacks for SF collection [22]. Gardiner et al. found that an ability to provide a SF specimen on request indicated a life expectancy for PCa patients comparable to cancer-free age-matched peers, i.e., younger subjects [7]. Our results seem to strongly confirm this observation. In the current population, age was an independent predictor of semen collection; younger patients accepted more frequently to participate and were able to collect then older ones. Moreover, our case series shows that patients with high levels of PSA, ISUP, and a higher percentage of cancer were more likely to participate in the study. In our study age was confirmed to be an independent predictor of sampling. Several analyses showed a high prevalence of ejaculatory dysfunction among patients with cancer and cancer survivors [23]. In our series, we found that 60% of men who had accepted to participate and were not able to collect their SF reported a psychogenic inhibition; all our patients had received a diagnosis of PCa. It is not unusual to feel anxious and depressed following a cancer diagnosis and waiting for a radical treatment or specific therapies [24]. PCa diagnosis and the fear of “specific” effects of the treatments may be the main cause of physiological changes that affected SF collection in our population. Another drawback for SF collection is related to men with erectile dysfunction. In our cohort, 15% of patients suffered of ED and failed to get SF (Figure 1). Evidence has highlighted the influence of alpha1-adrenoceptor antagonists on ejaculatory function [25,26,27]. Most of patients under alpha blockers cannot provide pure SF due to retrograde ejaculation. In our population, 24.8% of patients were not able to collect their SF due to alpha blocker therapy. Finally, a major issue is that patients may not be inclined to provide SF due to personal, social, religious, or ethical issues [28,29]. This is a wide framework, which was not investigated in patients who rejected because it was considered un-ethical investigating their personal choices. Lower urinary tract symptoms (LUTS) were not included in our analysis although they represent an important determinant of anxiety and distress in men and may impact on ejaculatory function.

Another critical point to assess for the implementation of SF liquid biopsy is the actual isolation and analysis of cancer-derived cells. Thus far, CTC isolation from semen has been performed in rather small cohorts of patients (maximum 15 individuals per study) and with different approaches, making it difficult to draw some generalizations [7,8,9,10]. Here, we applied a multi-color FACS approach, conceptually similar to the one used by Barren and colleagues on a much larger cohort of PCa patients. However, we used a different combination of markers for cell sorting, with the aim of selecting a more specific sub-population of cells. For instance, we included markers to identify dead cells and immune cells, and we did not use PSA, as it was reported to be less specific than PSMA in labelling prostate-derived cells [8]. EpCAM, whose overexpression has been linked to PCA progression, was used to label and characterize cancer-derived cells but was not used for sorting, as we could not exclude the presence of PCa cells expressing low levels of this marker. Indeed, our analyses confirmed that the estimated EpCAM^high^ cell fraction (P3/P2) does not correlate with clinical parameters, apart from age, and does not seem to add specificity to the PSMA labeling (P2 fraction) in PCa detection. It is interesting to note that, despite the differences between our protocol and that of Barren and colleagues, some findings are consistent: first, the number of isolated cells is generally high, and largely variable from sample to sample; and second, the detection/enumeration of prostate cells in semen alone does not provide useful clinical information [8]. Variability in particular seems to currently represent a major issue in the perspective of further developing the method for clinical application. Strategies to reduce such variability would probably need to consider various aspects including: (1) further streamline the protocol to reduce variability in the collection and pre-processing of the samples (indeed, self-sampling is inherently less controlled than a medical procedure performed by a professional); (2) find approaches to correct for intra-patient heterogeneity in cell populations released in semen/urine; and (3) identify additional and/or more selective markers for detecting PCa cells among prostate-derived cells. For instance, cytokeratins and glypican-1 were recently used to identify PCa-derived cells in semen after microfluidic isolation. These two markers seem to increase the specificity of detection compared to PSMA alone, as they allowed isolation of CTC from all 15 PCa patients analyzed, but not from most healthy donors (13/15) [10]. Moreover, isolated cells could be characterized using potentially prognostic markers, such as the tetraspanin CD82 (also known as KAI1), which suppresses metastasis in a variety of solid tumors, including PCa. Generally, maintained expression of CD82 in primary tumors correlates with a better prognosis compared to tumors with reduced or missing expression of CD82. For primary PCa, CD82 might represent both an early diagnostic and a prognostic marker because its expression is inversely correlated with GS (being absent in patients with GS ≥ 7), and the stratification of PCa patients based on CD82 expression in blood CTCs allowed the identification of patients with reduced survival and a higher risk of biochemical recurrence [30]. 

Finally, as CTC isolation was performed in parallel on both SF and urine, we present, for the first time, a side-to-side comparison of the two bio-fluids as a source of prostate-derived cells. Indeed, we were able to observe, under the same analytical conditions, a significantly higher number of prostate-derived cells in SF than urine. Other than that, the obtained results were mostly comparable in the two fluids, with a consistent positive correlation between patient age and the fraction of EpCAM^high^ cells over prostate-enriched cells (PSMA+). Nonetheless, more detailed molecular analyses on the isolated cells will be needed to detect patient-specific cancer lesions to help patient stratification and guide treatment.

## 5. Conclusions

Establishing SF-based non-invasive diagnostic methods could provide an alternative way for the early diagnosis and prognosis of PCa, while also facilitating monitoring of disease progression and treatment response in case of active surveillance or focal therapy [31]. A SF-based test for PCa will only be used in the clinic if it can reliably outperform or complement the current diagnostic pathways. However, the transition from theory to practice will not be straightforward. Our study provides evidence of limited acceptance and yield of SF collection among PCa patients, suggesting that liquid biopsy of SF might be better suited for younger patients, possibly in the context of early PCa detection or treatment stratification. In addition, technical optimization of protocols to isolate CTCs, either by cytofluorimetry (or other marker-dependent methods) or microfluidics/label-free strategies, is still needed to make these approaches clinically relevant. Once these issues are addressed, we are confident that SF-derived biomarkers can contribute to the diagnostic and prognostic armamentarium for PCa in the future.

## Figures and Tables

**Figure 1 cancers-15-02666-f001:**
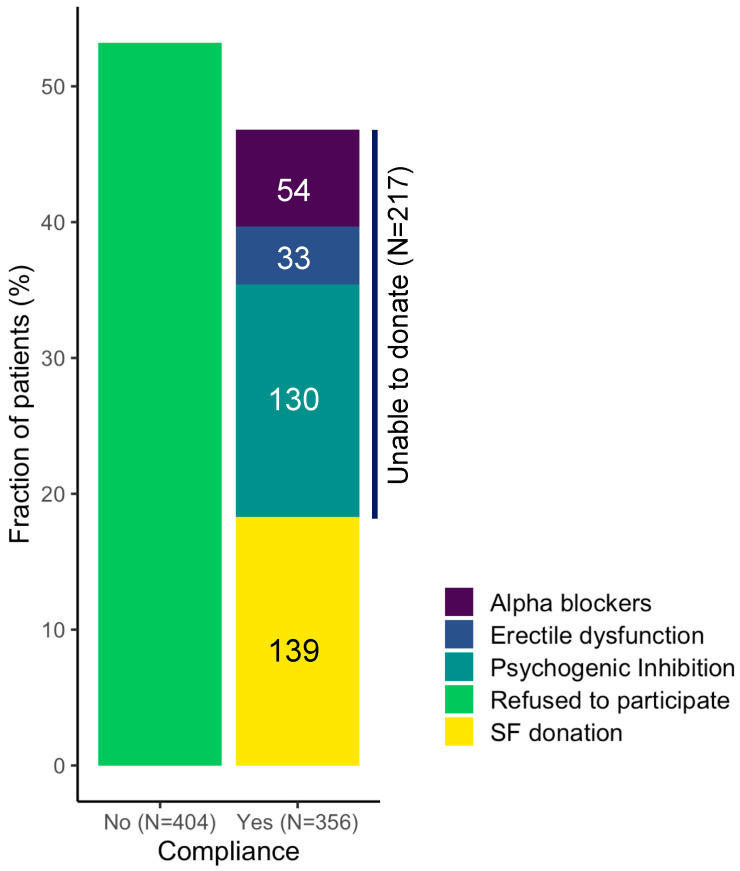
Compliance to the study and semen self-sampling yield. The bar plot shows the percentage of patients that declined (**left**) or accepted (**right**) to participate in the study. Among the patients who accepted, the percentage of patients that were able or not to provide a sample of ejaculate is indicated, with the reason for the lack of donation. Numbers of patients are also reported.

**Figure 2 cancers-15-02666-f002:**
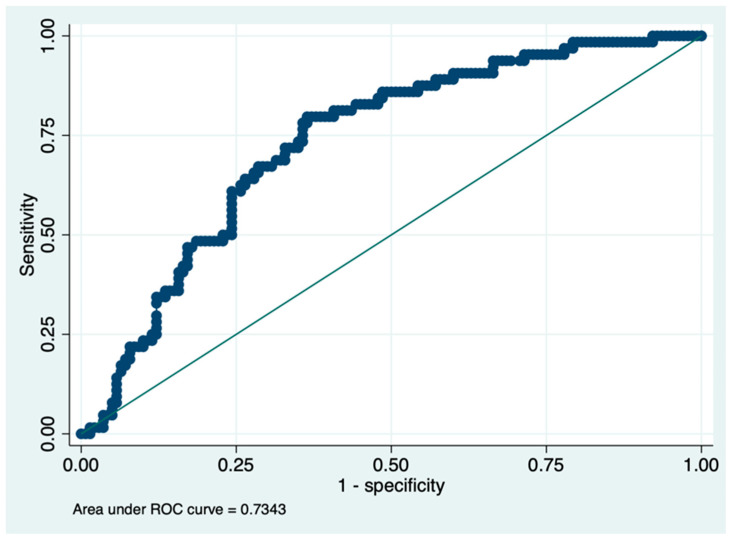
ROC analysis for semen collection.

**Figure 3 cancers-15-02666-f003:**
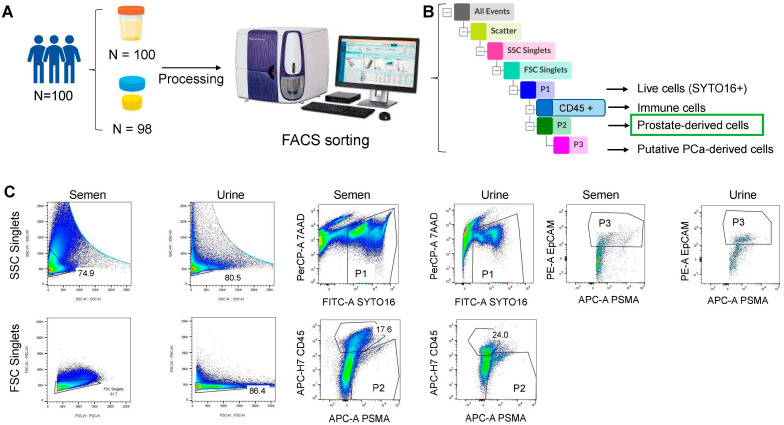
Multi-color FACS protocol for the isolation of prostate-derived cells from SF and urine, with the gate strategy applied. (**A**) Schematic representation of the experimental strategy. (**B**) Schematic representation of the population hierarchy. (**C**) Events characterization of one illustrative patient, in both seminal fluid and urine. Percentages indicated on plots are the fraction of proper gate on parent population and are automatically calculated by the software. The colors of clouds depend on the density of occurring events. The analysis and plots layout were created using FlowJo v10.8.1.

**Figure 4 cancers-15-02666-f004:**
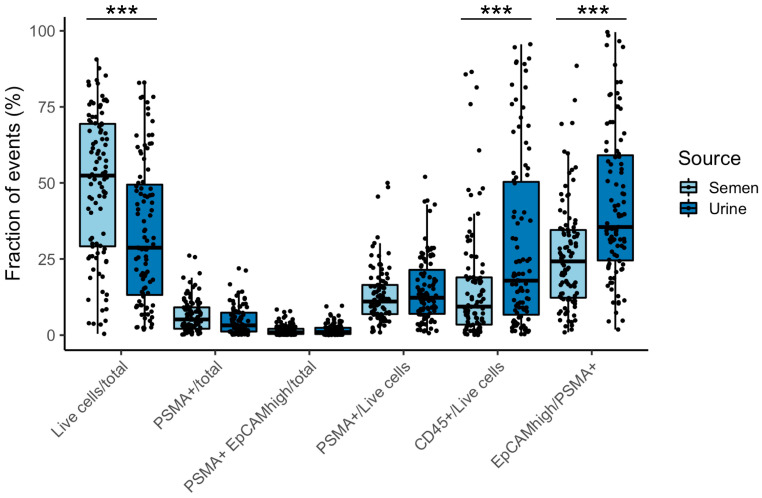
Enumeration of cell subpopulations analyzed by FACS from semen and urine. The number of cells is expressed as the percentage of events recorded in each gate out of the total or of the gated-parent population. The boxplots show the comparison between semen (blue) and urine (green) subpopulations in the analyzed patients. Significance was calculated by the using Mann–Whitney statistical test (*** *p*-value < 0.001). The data used to generate the Figure are available in Appendix A.

**Figure 5 cancers-15-02666-f005:**
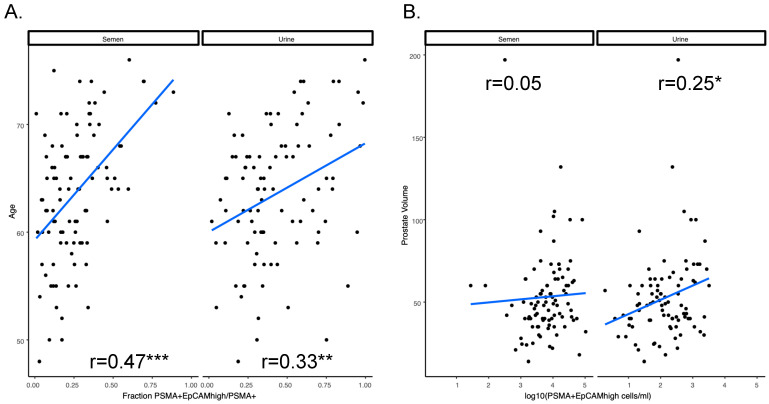
Correlation of selected clinical parameters with cancer-derived cells isolated from semen and urine. (**A**) Patient age positively correlates with the fraction of EpCAM^high^ cells over prostate-enriched cells (PSMA+) in both semen and urine. (**B**) Prostate volume positively correlates with the concentration (cells/mL) of EpCAM^high^ cells in urine only. Significance was calculated by using one-way ANOVA test (* *p*-value < 0.05, ** *p*-value < 0.01, *** *p*-value < 0.001). The data used to generate the Figure are available in Appendix A.

**Table 1 cancers-15-02666-t001:** Multi-color fluorescent dyes used for FACS analysis.

Fluorescent Dye	Target	Fluorophore	Concentration	Quantity per Sample (μL)	Dilution
7-AAD	Apoptotic cells	PerCP	500 μg/mL	10	1:1
Syto16	Live cells	FITC	1 mM solution in DMSO	8	1:10
CD45	Immune cells	APC-H7	200 μg/mL	2.5	1:1
PSMA	Prostate cells	APC	160 μg/mL	1.25	1:1
EpCAM	Epithelial cells	PE	3 μg/mL	1.25	1:1

7-AAD, 7-Aminoactinomycin D; CD45, Leukocyte Common Antigen; PSMA, Prostate Membrane Specific Antigen; EpCAM, Epithelial Cell Adhesion Molecule; PerCP, Peridinin-Chlorophyll-protein; FITC, Fluorescein isothiocyanate; APC, Allophycocyanin; PE, phycoerythrin.

**Table 2 cancers-15-02666-t002:** Comparison among patients’ demographic and clinical characteristics (overall, enrolled patients, and patients who declined participation).

		Total (*n* = 760)	Declined (*n* = 404)	Accepted (*n* = 356)	*p*-Value
Age (years), median (IQR)		66 (60–71)	66 (60–71)	65 (59–71)	0.48 ^§^
History of PCa, *n* (%)	No	675 (88.8%)	351 (86.9%)	324 (91.0%)	0.071 *
	Yes	85 (11.2%)	53 (13.1%)	32 (9.0%)	
Pre-biopsy PSA (ng/mL), median (IQR)		7.01 (5.11–11.06)	6.795 (5–10.21)	7.675 (5.2–12.4)	**0.016 ^§^**
PIRADS, *n* (%)	1	33 (4.3%)	29 (7.2%)	4 (1.1%)	**<0.001 ***
	2	1 (0.1%)	1 (0.2%)	0 (0.0%)	
	3	95 (12.5%)	65 (16.1%)	30 (8.4%)	
	4	201 (26.4%)	109 (27.0%)	92 (25.8%)	
	5	80 (10.5%)	23 (5.7%)	57 (16.0%)	
	N.A.	350 (46.1%)	177 (43.8%)	173 (48.6%)	
Prostate volume at MRI (cc), median (IQR)		50 (36–70)	56 (42–74)	48 (35–66)	**0.014 ^§^**
ISUP at final histological report, *n* (%)	1	16 (2.1%)	11 (2.7%)	5 (1.4%)	**0.004 ***
	2	127 (16.7%)	53 (13.1%)	74 (20.8%)	
	3	78 (10.3%)	23 (5.7%)	55 (15.4%)	
	4	22 (2.9%)	4 (1.0%)	18 (5.1%)	
	5	36 (4.7%)	9 (2.2%)	27 (7.6%)	
	N.A.	481 (63.3%)	304 (75.2%)	177 (49.7%)	
pT, *n* (%)	2a	29 (3.8%)	10 (2.5%)	19 (5.3%)	0.21 *
	2b	2 (0.3%)	1 (0.2%)	1 (0.3%)	
	2c	148 (19.5%)	62 (15.3%)	86 (24.2%)	
	3a	71 (9.3%)	21 (5.2%)	50 (14.0%)	
	3b	29 (3.8%)	6 (1.5%)	23 (6.5%)	
	N.A.	481 (63.3%)	304 (75.2%)	177 (49.7%)	
pN, *n* (%)	0	199 (26.2%)	77 (19.1%)	122 (34.3%)	0.076 *
	1	28 (3.7%)	6 (1.5%)	22 (6.2%)	
	N.A.	533 (70.1%)	321 (79.5%)	212 (59.6%)	
Positive margins (R), *n* (%)	0	228 (30.0%)	85 (21.0%)	143 (40.2%)	0.29 *
	1	51 (6.7%)	15 (3.7%)	36 (10.1%)	
	N.A.	481 (63.3%)	304 (75.2%)	177 (49.7%)	
% of tumor, median (IQR)		15 (8–20)	10 (5–20)	15 (10–25)	**0.017 ^§^**
EPE, *n* (%)	No	211 (27.8%)	78 (19.3%)	133 (37.4%)	0.49 *
	Yes	68 (8.9%)	22 (5.4%)	46 (12.9%)	
	N.A.	481 (63.3%)	304 (75.2%)	177 (49.7%)	

^§^ Wilcoxon rank-sum test; * Pearson’s chi-squared test; EPE: extra-prostatic extension; ISUP: International Society of Urological Pathology; PCa: prostate cancer; IQR: interquartile range; PIRADS: Prostate Imaging Reporting And Data System; MRI: magnetic resonance imaging; N.A.: not available. Parameters for which a statistically significant difference is present are bolded.

**Table 3 cancers-15-02666-t003:** Comparison among demographic and clinical characteristics of patients that were able to donate an ejaculate sample and those who were not.

		Semen Not Collected (*n* = 217)	Semen Collected (*n* = 139)	*p*-Value
Age (years), median (IQR)		68 (61–73)	64 (58.5–68)	**<0.001 ^§^**
History of PCa, *n* (%)	No	196 (90.3%)	128 (92.1%)	0.57 *
	Yes	21 (9.7%)	11 (7.9%)	
Pre-biopsy PSA (ng/mL), median (IQR)		8.14 (5.3–14.47)	7.08 (5.13–10.48)	0.094 ^§^
PIRADS, *n* (%)	1	4 (1.8%)	0 (0.0%)	0.42 *
	3	21 (9.7%)	9 (6.5%)	
	4	60 (27.6%)	32 (23.0%)	
	5	35 (16.1%)	22 (15.8%)	
	N.A.	97 (44.7%)	76 (54.7%)	
Prostate volume at MRI (cc), median (IQR)		55 (59–82)	40 (32–50)	**<0.001 ^§^**
ISUP at final histological report, *n* (%)	1	3 (1.4%)	2 (1.4%)	0.16 *
	2	17 (7.8%)	57 (41.0%)	
	3	20 (9.2%)	35 (25.2%)	
	4	3 (1.4%)	15 (10.8%)	
	5	7 (3.2%)	20 (14.4%)	
	N.A.	167 (77.0%)	10 (7.2%)	
pT, *n* (%)	2a	6 (2.8%)	13 (9.4%)	0.89 *
	2b	0 (0.0%)	1 (0.7%)	
	2c	22 (10.1%)	64 (46.0%)	
	3a	16 (7.4%)	34 (24.5%)	
	3b	6 (2.8%)	17 (12.2%)	
	N.A.	167 (77.0%)	10 (7.2%)	
pN, *n* (%)	0	37 (17.1%)	85 (61.2%)	0.77 *
	1	6 (2.8%)	16 (11.5%)	
	N.A.	174 (80.2%)	38 (27.3%)	
Positive margins (R), *n* (%)	0	42 (19.4%)	101 (72.7%)	0.39 *
	1	8 (3.7%)	28 (20.1%)	
	N.A.	167 (77.0%)	10 (7.2%)	
% of tumor, median (IQR)		20 (8–30)	15 (10–25)	0.24 ^§^
EPE, *n* (%)	No	37 (17.1%)	96 (69.1%)	0.95 *
	Yes	13 (6.0%)	33 (23.7%)	
	N.A.	167 (77.0%)	10 (7.2%)	

^§^ Wilcoxon rank-sum test; * Pearson’s chi-squared test; EPE: extra-prostatic extension; ISUP: International Society of Urological Pathology; PCa: prostate cancer; IQR: interquartile range; PIRADS: Prostate Imaging Reporting And Data System; MRI: magnetic resonance imaging; N.A.: not available. Parameters for which a statistically significant difference is present are bolded.

**Table 4 cancers-15-02666-t004:** Univariate and multivariate analysis for semen collection.

	Univariate				Multivariate			
	OR	z	*p*	[95% CI]	OR	z	*p*	[95% CI]
Age	0.95	−4.28	**<0.001**	0.92–0.97	0.95	−1.94	**0.052**	0.91–1.0
Prostate Volume	0.96	−3.96	**<0.001**	0.95–0.098	0.97	−3.64	**0.001**	0.95–0.98
PSA pre-biopsy	1	−1.23	0.217	0.97–1	-			
PIRADS	1.9	3.73	**<0.001**	1.37–2.38	-			

CI: confidence interval; OR: odds ratio; PIRADS: Prostate Imaging Reporting And Data System. Parameters for which a statistically significant difference is present are bolded.

## Data Availability

All the data presented in this study are available within the article and Appendix A.

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
