# Peer review of "Evaluation of Semen Self-Sampling Yield Predictors and CTC Isolation by Multi-Color Flow Cytometry for Liquid Biopsy of Localized Prostate Cancer"

_cancers, 2023, doi:10.3390/cancers15102666_

Round 1

Reviewer 1 Report (Previous Reviewer 3)

The authors have significantly improved the manuscript both in content in presentation and have addressed all previous issues.

Some minor points were found in current version:

344-345 “In case of urine, the sample volume is not a limiting factor, and the storage and transport of the sample are easy[19].”
Probably wrong reference number.

Table 4 – Prostate volume, multivariable – p value is exactly 0, probably missing digits.

Some minor typos are present throughout the manuscript – for example line 333 “to non-metastatic diseas[16]” or Table 4 “Univariate e Multivariable”.

Author Response

Some minor points were found in current version:

Point 1. 344-345 “In case of urine, the sample volume is not a limiting factor, and the storage and transport of the sample are easy[19].” Probably wrong reference number.

Response: We thank the reviewer for noticing. We have now inserted the right reference number [17].

Point2. Table 4 – Prostate volume, multivariable – p value is exactly 0, probably missing digits.

Response: Thank you for bringing to our attention the missing value in Table 4. We have now inserted the corrected p value (<0.001).

Point3. Some minor typos are present throughout the manuscript – for example line 333 “to non-metastatic diseas[16]” or Table 4 “Univariate e Multivariable”.

Response: The entire manuscript was carefully re-read to amend typos.

Reviewer 2 Report (Previous Reviewer 2)

The authors have suitably addressed all the concerns I raised about this paper. So I think this manuscript is ready for publication.

Author Response

Response: No further revision requested by the reviewer.

Reviewer 3 Report (Previous Reviewer 1)

This revised manuscript explored the feasibility of using self-collected seminal fluid as sample for detection of CTCs in a cohort of patient diagnosed with localized prostate cancer. It offered insights using real-world experience that the challenges in collecting samples in this patient population. However, authors did show that CTCs can be readily detected from SF and post ejaculation urine. The data were well presented, and the manuscript follows a logical flow. The conclusion is supported by the results presented. I appreciate the efforts from authors to improve the quality of the manuscript through revision and have no additional comments. 

Author Response

Response: No further revision requested by the reviewer.

Reviewer 4 Report (New Reviewer)

Summary:

Circulating tumor cells (CTCs) and circulating cell-free tumor nucleic acids (ctNAs) in peripheral circulation are alternatives to conventional tissue biopsy in the management of advanced prostate cancer. However, they have shown poor performance in clinically localized prostate cancer. Considering the longstanding evidence that seminal fluid and urine might contain prostate cancer cells, they could be ideal liquid for a non-invasive early diagnosis of prostate cancer. Cesare Saitta, et al identified the main favorable predictor for semen collection by assessing the correlation between ejaculation capability and clinical characteristics of patients with localized prostate cancer, isolated prostate-derived cells as well as prostate cancer cells from seminal fluid and urine by multi-color flow cytometry, and evaluated the diagnostic and prognostic applicability of the isolated cells by correlating them with clinical parameters of patients with localized prostate cancer. However, the statistical analyses indicate that cells in seminal fluid and urine fail to guide prostate cancer diagnosis and prognosis.

General comments:

(1) Insufficient introduction of clinical characteristics of prostate cancer. For example, in Table 2, the PIRADS is divided into 6 groups: 1, 2, 3, 4, 5, and NA. What does each group indicate in terms of the characteristics of prostate cancer? Please provide sufficient introduction of PIRADS, ISUP, pT, pN, and positive margins.

(2) Putative prostate cancer cell number largely affects the results of the evaluation of diagnostic applicability of seminal fluid and urine (Figure 5 (a) and (b)), which requires the authors’ careful identification and enumeration of SYTO16+PSMA+CD45-EpCAMhigh cells. Did the authors use positive controls, such as prostate cancer cell lines, to determine the gate of EpCAMhigh cells? Did the authors perform validation of collected cells following the flow cytometry?

(3) The authors obtained putative prostate cancer cells from healthy donors. Is this a false positive due to the isolation method? If it is so, would the false positive rate grow larger in the patient donors as the total number of cells in seminal fluid and urine is much larger?

Author Response

General comments:

(1) Insufficient introduction of clinical characteristics of prostate cancer. For example, in Table 2, the PIRADS is divided into 6 groups: 1, 2, 3, 4, 5, and NA. What does each group indicate in terms of the characteristics of prostate cancer? Please provide sufficient introduction of PIRADS, ISUP, pT, pN, and positive margins.

Response: Thank you for your valuable feedback. We have carefully considered your suggestions and have inserted a more detailed description of the clinical characteristics of prostate cancer (PIRADS, ISUP, pT, pN, and positive margins), providing a more comprehensive overview of our data (see Results lines 190-220). Additionally, we have updated Table 2 to reflect the correct classification of patients. In this regard we acknowledge a mistake regarding the pT stage (one patient was previously classified as pT1 instead of pT2a). We apologize for any confusion or inconvenience caused by this error. We hope that these changes have improved the clarity and quality of our manuscript.

(2) Putative prostate cancer cell number largely affects the results of the evaluation of diagnostic applicability of seminal fluid and urine (Figure 5 (a) and (b)), which requires the authors’ careful identification and enumeration of SYTO16+PSMA+CD45-EpCAMhigh cells. Did the authors use positive controls, such as prostate cancer cell lines, to determine the gate of EpCAMhigh cells? Did the authors perform validation of collected cells following the flow cytometry?

Response: Yes, to set up the protocol and define the gates for flow cytometry we used positive controls (LNCaP cells) and validation of collected cells, as described in Lazzeri et al., 2018 (doi.org/10.1016/j.juro.2018.02.430). Briefly, semen samples from healthy donors were spiked with different amount (10, 100, 1000) of LNCaP cells, a human prostate adenocarcinoma cell line with a well-defined 1-bp insertion within exon 9 of the JAK1 gene (NM_002227.3: c.1282_1283insC). After cell-sorting, we were than able to detect the presence of LNCaP cells by fluorescent PCR. The estimated recovery of LNCaP cells was 70.4% ± 10.8%. We have now added a Supplementary Figure (Supplementary Figure S1, Methods lines 129-134) with further details on these control experiments.

(3) The authors obtained putative prostate cancer cells from healthy donors. Is this a false positive due to the isolation method? If it is so, would the false positive rate grow larger in the patient donors as the total number of cells in seminal fluid and urine is much larger?

Response: The presence of putative prostate cancer cells in healthy donors is likely due to the fact that the combination of selected markers is not 100% selective and specific for PCa, as already mentioned in the Discussion (lines 406-411). It is also possible -as pointed out by the Reviewer- that in PCa patients a non-negligible proportion of isolated cells would correspond to false positives (non-cancerous cells). That is why in our Discussion we already stressed the need of addressing this type of issue in various ways (Discussion section, lines 436-441 and 450-456), including: 1) find approaches to correct for intra-patient heterogeneity in cell populations released in semen/urine; 2) identify additional and/or more selective markers for detecting PCa cells among prostate-derived cells; 3) perform more detailed molecular analyses on the isolated cells of each patient, for example by looking for tumor-specific genetic lesions or altered methylation patterns.

Round 2

Reviewer 4 Report (New Reviewer)

The authors have addressed my previous concerns and questions. I believe that the manuscript has been sufficiently improved to warrant publication in Cancers.

This manuscript is a resubmission of an earlier submission. The following is a list of the peer review reports and author responses from that submission.

Round 1

Reviewer 1 Report

The manuscript by Saitta et.al. explored the possibility of liquid biopsy for early detection of prostate cancer by measuring prostate cancer circulating tumor cells from seminal fluid and urine. Despite the technical feasibility of detecting CTCs from the samples, only about one fifth of patients in the cohort were able to successfully provide seminal fluid for analysis, and there was no correlation between CTC numbers and clinical outcome. One major caveat is that patients included in the study were already diagnosed with cancer and scheduled for definitive treatment, so it is not the appropriate cohort to test if LB from seminal fluid is useful in the context of early prostate cancer detection. It remains unknown if patient will have a higher participation rate and success rate to provide semen samples without knowing the established diagnosis of prostate cancer, which is the real questions the authors set to answer. Therefore, the major conclusion "LB of SF is feasible ... in the context of early PCa detection" is unfounded and not supported by the data presented. 

Reviewer 2 Report

In this study the authors report a novel method of detection of Prostate Cancer derived cells from a sizable group of Patient Cohort from Seminal Fluid which could serve as a potential Biomarkers. This technique could aid in early diagnosis. This study seems important for the field. However, I had the following concerns about this study.

1. The advantages of this technique over collecting urine/semen seem to be limited and unclear. They need to make some more arguments in this regard.

2. They fail to state the "Stage" of Prostate Cancer in the patients that they collected data from and whether the levels of bio-markers correlates to that.

3. They state that half the patients did not consent to sample collections. How can awareness be raised to ensure that more patients volunteer to do that in the future ?

4. They fail to mention what kind of  additional/new SF-derived biomarkers that they will use in future studies to same this procedure better than other methods.

5. As long as they don't  get rid of variability and report any co-relation to cl clinical parameters it might be difficult to market this strategy further.

Reviewer 3 Report

While the topics discussed in the article are of interest, the manuscript itself is in too rough state and requires improvement out of the scope of major revision, hence I recommend to authors to withdraw the manuscript and resubmit it after addressing significant issues present.

The “early detection” part in title is misleading, as the study was performed only in patients already subjected to biopsy and diagnosed with PCa, meaning these patients have already reached the clinically detectable disease stage that warranted aforementioned interventions.

The rest of the title should be rephrased, as currently it can be understood as exploration of CTC isolation predictors.

As simple summary states that the primary objective was the evaluation of seminal fluid sampling yield it should be also primary part of the title.

As only patients with diagnosed localized PCa participated in the study, it limits the applicability of resultsa as for early diagnostics due to lack of healthy controls both for SF yield study and prostate CTC background level, but also for disease monitoring, as there was no post-surgery and PCa progression groups.

The simple summary and abstract contradict each other – one stating that “only one third of patients… accepted and were able to collect” and other that “almost half not consenting” and “collection yield was 18.2%”.

In introduction, line 68 “Besides few attempts to isolate CTCs from semen 2018 we showed preliminary results indicating that PCa cells can be retrieved from semen and cancer-specific genetic alterations may be efficiently detected starting from a heterogeneous cell 70 population[7], [8].” – the sentence suggests authors referencing their previous work but as [7],[8] point at earlier results by other collectives, no such work is referenced, and “2018” is just inserted in the middle of the sentence.

In line 152 the comparison of pre- and post-surgery values are mentioned, but no such are found in the manuscript.

In lines 196-200, logistic regression model and uni- and multivariate analysis are mentioned, but only results presented are two predictors reaching significance and ROC curve in Figure 2.
As these results are main topic of the manuscript, whole data should be shown – both model and results for every predictor tested.
Individual data for samples is not presented.

In lines 186-188 its mentioned that 139 patients were able to collect SF (and, presumably, post-ejaculatory urine?), but only 71 had urine, blood and tumor specimen obtained, and later, in line 220, usage of 100 SF samples is mentioned – if analysis was done not only on samples from patients with all specimens obtained, why whole set of 139 samples was not used?  Also, if the blood was also collected, why no comparison was done on CTC detection possibility between SF/urine and blood, considering that expected advantages in CTC yield for SF were discussed in introduction and discussion sections?

Figure 4 is clearly unfinished, specifically the legend and labels.

Same as for SF yield predictors, the data for correlation between SF/urine results and clinical parameters should be presented.

Table 2 contains only data for groups that accepted or declined participation, without any difference for age between them, but claim that age is a predictor for SF collection. Authors should show same data for subgroups among participated patients that were able or not to collect SF, and data for age subgroups.
Also, its not clear what exactly p values represent for such parameters as PIRADS or ISUP as two groups with multiple rows are presented but only one p value per parameter.

Simple summary in line 18 mentions “while the main obstacles were living far from the Hospital, lower educational levels…” but no data concerning those is presented anywhere.

Parts of results sections are essentially duplicate – compare lines 182-187 and 192-195 with lines 200-205.

The references 25 and 26 are not mentioned anywhere in the manuscript.

References 28,29,30 and 31 seem to be added specifically for authors self-citations, as PET/CT is not the subject of presented work.

Also it should be noted that there are parts in discussion section almost word by word repeating previous work (M. J. Roberts, R. S. Richards, R. A. Gardiner, e L. A. Selth, «Seminal fluid: a useful source of prostate cancer biomarkers?», Biomark. Med., vol. 9, fasc. 2, pp. 77–80, feb. 2015, doi: 10.2217/bmm.14.110. 447, cited in current manuscript under [19]) – for example:

lines 286-288 in current manuscript

“Indeed, prostatic constituents are highly enriched in SF compared to other bodily fluids; PSA was  originally discovered in SF, where it exists at a concentration significantly higher than in blood serum [15]. Unlike malignant prostatic epithelial cells, which would enter the circulation only following disruption of blood–tissue barriers, CTCs and cell-free nucleic acids are naturally released into SF by both normal and malignant glands. Finally, SF not 290 only contains cell-free material from the prostate but also cancer cells that might be 291 detectable prior to biopsy-based PCa diagnosis[7].”

compare with Roberts et al.
“First, prostatic constituents are highly enriched in SF compared with other bodily fluids. Indeed, PSA was originally discovered in SF, where it exists at a concentration approximately 5–6 orders of magnitude higher than in blood serum [3]. Second, unlike malignant prostatic epithelial cells and their products that only enter the circulation following transgression of blood–tissue barriers, cells and their secretions are released naturally into SF by both normal and malignant glands.  …… Third, SF not only contains cell-free material from the prostate but also cancer cells that are detectable prior to biopsy-based PCa diagnosis [4]”